# Oral Pyogenic Granuloma: A Narrative Review

**DOI:** 10.3390/ijms242316885

**Published:** 2023-11-28

**Authors:** Sarah Monserrat Lomeli Martinez, Nadia Guadalupe Carrillo Contreras, Juan Ramón Gómez Sandoval, José Sergio Zepeda Nuño, Juan Carlos Gomez Mireles, Juan José Varela Hernández, Ana Esther Mercado-González, Rubén Alberto Bayardo González, Adrián Fernando Gutiérrez-Maldonado

**Affiliations:** 1Department of Medical and Life Sciences, University of Guadalajara (CUCiénega-UdeG), 1115 Ave. Universidad, Ocotlán 47810, Jalisco, Mexico; sarah.lomeli@academicos.udg.mx (S.M.L.M.); juan.varela@academicos.udg.mx (J.J.V.H.); 2Master of Public Health, Department of Wellbeing and Sustainable Development, University of Guadalajara (CUNorte-UdeG), 23 Federal Highway, Km. 191, Colotlán 46200, Jalisco, Mexico; 3Periodontics Program, Department of Integrated Dentistry Clinics, University of Guadalajara (CUCS-UdeG), 950 Sierra Mojada, Guadalajara 44340, Jalisco, Mexico; nadia.carrillo2999@alumnos.udg.mx (N.G.C.C.); juan.ramongom@academicos.udg.mx (J.R.G.S.); carlos.gomezm@academicos.udg.mx (J.C.G.M.); 4Prostodontics Program, Department of Integrated Dentistry Clinics, University of Guadalajara (CUCS-UdeG), 950 Sierra Mojada, Guadalajara 44340, Jalisco, Mexico; 5Research Institute of Dentistry, Department of Integrated Dentistry Clinics, University of Guadalajara (CUCS-UdeG), 950 Sierra Mojada, Guadalajara 44340, Jalisco, Mexico; 6Microbiology and Pathology Department, Pathology Laboratory, University of Guadalajara (CUCS-UdeG), 950 Sierra Mojada, Guadalajara 44340, Jalisco, Mexico; jsergio.zepeda@academicos.udg.mx; 7Antiguo Hospital Civil de Guadalajara “Fray Antonio Alcalde”, 777 Coronel Calderón, Guadalajara 44200, Jalisco, Mexico; anamer62@gmail.com; 8Department of Integrated Dentistry Clinics, University of Guadalajara (CUCS-UdeG), 950 Sierra Mojada, Guadalajara 44340, Jalisco, Mexico; ruben.bayardo@cucs.udg.mx

**Keywords:** pyogenic granuloma, hyperplastic lesion, lobulated capillary hemangioma

## Abstract

Pyogenic granuloma (PG) is a benign vascular lesion found predominantly in the oral cavity. Characterized by rapid growth and propensity to bleed, PG presents diagnostic challenges due to its similarity and alarming proliferation. This narrative review synthesizes current knowledge on the epidemiology, etiopathogenesis, clinical manifestations, and management of oral PG, with emphasis on recent advances in diagnostic and therapeutic approaches. The epidemiology of the injury is meticulously analyzed, revealing a higher incidence in women and a wide range of ages of onset. It delves into the etiopathogenesis, highlighting the uncertainty surrounding the exact causal factors, although historical attributions suggest an infectious origin. It exhaustively analyzes the clinical and histopathological aspects of oral PG, offering information on its various presentations and the importance of an accurate diagnosis to guide effective treatment. It details treatment strategies, emphasizing the personalized approach based on individual patient characteristics. This comprehensive review consolidates current knowledge on oral PG, highlighting the need for further research to clarify its pathogenesis and optimize treatment protocols.

## 1. Introduction

Pyogenic granuloma (PG) is a benign connective tissue proliferation that is predominantly characterized by granulation tissue hyperplasia, and it occurs frequently in the skin or mucous membranes [1,2,3,4]. It is considered one of the most common lesions responsible for soft tissue enlargements, due to its rapid and alarming growth rate [1,5,6,7].

Pyogenic granuloma (PG) was first described in 1897 by Poncet and Dor, who reported four patients with “vascular tumors” on the fingers which they named “*Botrichomycosis hominis*” [8]. The term “pyogenic granuloma” was introduced in 1904 by Hartzell. However, the name is considered inappropriate as it is neither related to pus formation, nor is it histologically a true granuloma [2,9,10]. Due to the controversy regarding its true pathological nature, this lesion has been given several names such as granuloma pediculatum benignum, benign vascular tumor, septic granuloma, hemangiomatous granuloma, vascular epulis, fibroangioma, polypoid capillary hemangioma, eruption capillary hemangioma, non-lobular capillary hemangioma, and Crocker and Hartzell’s disease [2]. In pregnant women, GP is identified as pregnancy granuloma, pyogenic granuloma of pregnancy, or granuloma gravidarum [11]. In the dermatological literature, Cawson et al. (1998) have described this disease as “granuloma telangiectacticum” due to the presence of numerous blood vessels observed in histological sections. Some pyogenic granulomas (also known as lobular capillary hemangiomas) are categorized as vascular tumors, according to the classification of the International Society for the Study of Vascular Anomalies (ISSVA, 2022).

Given that PG is a common lesion in the oral cavity, this article was aimed at reviewing its prevalence, etiopathogenesis, clinical picture, radiographical and histopathological features, as well as its differential diagnosis and treatment.

## 2. Epidemiology

Oral PG manifests within the range of 4.5 to 93 years, and it shows a preeminent incidence in the second and fifth decades, with the female gender often being more affected than the male gender [3,4]. On the other hand, some researchers have suggested that this disease manifests mostly in males younger than 18 years and in females aged 18–39 years, with equal gender distribution in older patients [10]. With regard to its prevalence, in a retrospective study conducted at the Oral and Maxillofacial Pathology Department, Faculty of Dentistry, Shiraz University of Medical Sciences in Iran, analysis was performed on 1000 biopsies of gingival lesions, out of which 92.4% of the cases were found to be non-neoplastic lesions. The most common reports were related to reactive lesions (71.8%), with the highest PG-associated prevalence of 24.6% [12,13]. In their descriptive study of 427 patients, PG accounted for 50.35% of all reactive oral cavity lesions reported at Dharwad Dental Institute in Karnataka, India. Shammin et al. [14] analyzed 244 cases of gingival lesions at the Department of Oral Pathology, Government Dental College, Calicut, India, and reported that non-neoplastic lesions comprised 75.5% of the cases, out of which PG accounted for 52.7%. A descriptive study of 923 Iranian pregnant women revealed that only 2 of them (0.22%) presented with PG [15]. On the other hand, a 4.2% PG prevalence has been reported in this group of patients in a medical center in Kerman/Iran [16]. These discrepancies in prevalence could be attributed to the differences in diagnostic methods applied. The diagnosis was based only on the clinical picture of the lesion, while Nejad et al. [15] implemented confirmation through histopathological studies.

## 3. Etiopathogenesis

Some factors are implicated in the etiopathogenesis of PG. However, the exact cause is unknown. Historically, some researchers consider it to be a pathology attributable to an infectious agent, hence the term “pyogenic” [3,17]. Kerr in 1951 was of the view that the factors that influence the progression of PG were bebotryomycosis, staphylococci, foreign particles, and the accumulation of infection in the endothelium of blood vessels [18]. In a study by Bhaskar and Jacoway, Gram-positive and Gram-negative bacilli were identified in GP. However, these microorganisms could be members of the oral microbiota, since they were more frequent in ulcerated lesions than in non-ulcerated lesions [19]. On the other hand, in 2001, Lee and Lynde pioneered the establishment of a statistically significant association between Bartonella seropositivity and PG, which has important therapeutic implications [17]. This association suggested the possibility of non-surgical treatment with an antibiotic such as erythromycin. In addition, they suggested that such treatment may also decrease recurrence and satellitosis of PG.

Several researchers define the PG as a “reactive” or “reparative” process. Regezi et al. consider PG a reactive or repairing process in which a certain stimulus generates an exuberant proliferation of connective tissue [2,3]. The etiological factors considered as stimuli that trigger this reactive process are trauma, dental calculus, dental biofilm, chronic irritation, pre-existing vascular lesions, chronic irritation due to exfoliation of primary teeth, injury of a primary tooth, eruption of permanent teeth, defective restorations in the area of the lesion, occlusal interference, food impaction, periodontitis, and trauma from toothbrushing [2,3,7]. These factors are summarized in Figure 1.

Pyogenic granuloma (PG) may manifest after a hypersensitivity reaction associated with the use of drugs such as calcineurin inhibitors (cyclosporine and tacrolimus), carbamazepine, phenytoin, nifedipine, levothyroxine and ramucirumab [20,21,22,23,24,25,26,27]. Additionally, PG is associated with retinoid, antineoplastic and antiretroviral agents [20]. The mechanism involved in the presence of PG following hematopoietic cell transplant (HCT) is unknown. Cheney and Lund described five cases of pediatric patients who developed oral PG after HCT, under treatment with the calcineurin inhibitors cyclosporine A or tacrolimus [21]. It has been suggested that a side effect of cyclosporine, which is less common in tacrolimus, is the proliferation of fibroblasts, thereby generating increased collagen synthesis in the oral mucosa, resulting in triggering of a critical gingival hyperplasia that leads to PG. Moreover, there is an increase in the level of connective tissue growth factor [22,28]. Cyclosporine decreases the production of collagenase and matrix metalloproteinases, and also upregulates the expressions of the inhibitors of these metalloproteinases, thereby playing a crucial role in tissue growth. These hyperplastic effects on the gums may be implicated in the development of PG associated with calcineurin inhibitors. Cheney et al. described a case series in which they included five pediatric/adolescent patients who developed oral PG after HCT for acute lymphoblastic leukemia, Fanconi anemia, nodular sclerosis Hodgkin’s lymphoma, or junctional epidermolysis bullosa [21]. It was suggested that calcineurin inhibitors, which are used for graft versus host disease, play a crucial role due to irritation and chronic inflammatory changes in the oral cavity, leading to tissue proliferation, and eventually to PG formation. Regarding carbamazepine, the release of angiogenic factors stimulated by the inflammatory process and the impairment of liver functions contribute to the development of PG [23]. Levothyroxine has been considered a possible etiological factor in PG through its pro-angiogenic and proliferative effects [24]. Piraccini et al. reported that a side effect in psoriasis or acne patients treated with retinoids (systemic isotretinoin, systemic etretinate, systemic acitretin, topical retinoic acid, and topical tazarotene) is the appearance of periungual PGs. Retinoids decrease the attachments amongst keratinocytes, exert angiogenic properties, and inhibit the activities of collagenases and gelatinases in vitro [20,29]. Antiretrovirals, particularly indinavir and lamivudine, are associated with the presence of PG in the nail folds, and the time of appearance varies from 2 months to 1 year from the initiation of therapy [20,25]. It has been suggested that protease inhibitors may have a retinoid-like effect by virtue of homologies between the amino acid sequences of cellular retinoic acid binding protein 1 (CRABP1) and the catalytic site of HIV-1 protease. Thus, the specific retinoid receptor may be occupied and activated by the antiretroviral agent, thereby increasing the activity of vitamin A and its analogues [30]. Multiple PGs have been described in metastatic carcinoma patients treated with antineoplastic drugs (epidermal growth factor receptor inhibitors, capecitabine, cyclosporin, docetaxel, and mitoxantrone) [20]. Aragaki et al. described two clinical cases with the presence of oral PG during administration of ramucirumab for gastric cancer. Ramucirumab, an entirely human IgG1 monoclonal antibody, favors the appearance of this pathology (PG) by generating systemic deterioration of the angiogenic balance and local deterioration of the oral environment [27].

Hormonal changes, especially in estrogen and progesterone during puberty and pregnancy, may promote the development of PG or pregnancy granuloma. Increased levels of these hormones during puberty deteriorate already established gingival inflammation by increasing the dilation and proliferation of blood vessels and releasing vasoactive mediators from damaged mast cells [2,31]. In pregnancy in particular, these hormonal changes have been associated with vascular, microbiological, cellular and immunological modifications which generate a favorable environment for the initiation and development of PG [11,31,32,33]. The hormones estrogen, progesterone and chorionic gonadotropin induce certain alterations in the microcirculatory system, including swelling of endothelial cells, increased adhesion of platelets and leukocytes to vessel walls, formation of microthrombi, disruption of perivascular mast cells, increased vascular permeability, and vascular proliferation [31,32] (Figure 2). The oral microbiota may present changes characterized by an increase in the proportion of anaerobic and aerobic bacteria such as *Bacteroides melaninogenicus*, *Prevotella intermedia*, and *Porphyromonas gingivalis*. Thus, high levels of *Fusobacterium nucleatum* and *Aggregatibacter actinomycetemcomitans* have also been observed, particularly in the second and third trimesters of pregnancy [11,31,32,34,35]. The cellular changes comprise a decrease in the keratinization of the gingival epithelium, an increase in epithelial glycogen, proliferation of fibroblasts, and a blockage in collagen degradation, leading to changes in the epithelial barrier that result in an increased response against irritating factors, especially dental biofilm [32]. Progesterone may act as an immunosuppressant in the periodontal tissues of pregnant women, thereby preventing the appearance of an acute inflammatory response to an irritant stimulus. However, it allows an increase in chronic tissue reactions, which results in an exaggerated appearance of inflammation clinically [36]. Additionally, there is a decrease in the antimicrobial activity of peripheral neutrophils which constitute essential components of the innate immune defenses of periodontal tissues. All these changes result in the exacerbation of the prevalence and/or severity of some pathologies in the oral cavity during pregnancy, particularly in the tissues, especially from the second month onwards, because it is exactly at this point that elevations in plasma estrogen and progesterone levels occur [31,33].

The imbalance between angiogenesis enhancers and inhibitors is one of the hypotheses for the etiopathogenesis of PG. It highlights the important role of certain factors such as basic fibroblast growth factor (bFGF), vascular endothelial growth factor (VEGF), tyrosine kinase with immunoglobulin-like, EGF-like domains-2 (Tie-2), angiopoietin-1 (Ang-1), angiopoietin-2 (Ang-2), ephrin-B2 and Eph-B4 in the processes involved in adult inflammatory neovascularization [37,38]. On the other hand, Shetty et al. reported that PG is triggered by the presence of local and/or systemic factors that generate the release of various endogenous substances (tumor cell angiogenic factors and vascular morphogenic factors), leading to alterations in the vascular system of the affected area [39]. Decorin is an integral component of new capillaries in in vivo angiogenesis, especially angiogenesis associated with severe inflammation. Nelimarkka et al. have demonstrated that decorin is present in the endothelial cells of capillary neovessels in PGs and in the granulation tissue of healing dermal wounds [40].

PG is also known as intravascular lobular capillary hemangioma (ILCH), as it is considered a benign vascular tumor that is usually located within the blood vessels of the head, neck and upper extremities [41,42,43]. In PG, excessive growth of vascular tissues is observed towards the interior of the vessel in addition to the invasion of surrounding tissues. In the histopathological analysis, it is observed that this type of hemangioma presents groups of capillaries enclosed by flattened endothelial cells within the upper layers of mucosa, as well as dense groups of vessels lined with fibrovascular endothelium with robust walls located in the deeper layers (Figure 2) [44]. Congenital endothelial hyperplasia is considered to be a contributing factor in its pathogenesis [45], and recent studies have also highlighted the participation of intussusceptive angiogenesis, also known as angiogenesis by division [46]. Histologically, ILCH exhibits a distinctive lobular architecture marked by a nonstratified endothelium that forms capillaries within an edematous fibromyxoid stroma, these lobes are separated by venular structures. Characteristic features, such as capillary-sized vessels and inter-endothelial contacts, distinguish ILCH from other vascular tumors [47,48].

PG is considered within the group of reactive oral lesions that develop in response to chronic irritation or trauma and where the inflammatory response of the tissue increases [49]. The chronic inflammatory immune response to chronic irritation is characterized by simultaneous tissue remodeling and repair. Remodeling begins with the migration of inflammatory cells regulating the immune system, the proliferation of vascular endothelial cells, fibroblasts and the synthesis of the extracellular matrix. Cytokines, growth factors and angiogenic factors play a definitive role in the repair process [50]. Mast cells and macrophages have been reported to contribute to the pathogenesis of reactive oral lesions through their effects on endothelial cells and fibroblasts. They synthesize increased amounts of basic Fibroblast Growth Factor (bFGF) to be released into the extracellular matrix and also promote neovascularization through pro-angiogenic cytokines primarily Vascular Endothelial Growth Factor (VEGF). Mast cell mediator, namely tryptase, promotes fibroblast activation, collagen deposition, and fibrosis [51,52,53,54]. Mast cells release mediators that increase vascular permeability and vasodilation, facilitating the migration of inflammatory cells [55]. The outcome of mutual mast cell–fibroblast interactions promotes granulation tissue formation, a hallmark of PG [55], which encouraged us to evaluate the mast cells numbers in both histopathological subtypes. Several authors [56], have found a positive correlation between the number of micro vessels and mast cells in a group of so-called reactive lesions, including LCH, fibrous hyperplasia, inflammatory fibrous hyperplasia, and peripheral giant cell lesions, while others found a negative correlation [57]. The major basic protein of eosinophils has proangiogenic effects and may participate in inflammation and tissue remodeling. Furthermore, in addition to VEGF, eosinophils have been shown to produce other proangiogenic factors such as IL-8 and nerve growth factor [58,59]. The contribution of eosinophils to the pathogenesis of oral reactive lesions was previously studied, but eosinophils were not found to be involved in the fibrotic process or in the variation of the microscopic characteristics of these lesions [54]. Mast cell mediator, namely tryptase, promotes fibroblast activation, collagen deposition, and fibrosis [51,52,53,54]. During the initial stage of the lesion, inflammatory cell infiltration, vascular proliferation, and appearance of myofibroblasts and type III collagen can be observed in response to the injury and repair. When the lesions mature, the connective tissue consists of densely packed type I collagen with no inflammation and the myofibroblasts disappear, suggestive of healing [60].

A study carried out by Vasanthi et al. in 2022 where they evaluated the severity of inflammation of reactive lesions, observed a great inflammatory activity in PG, followed by inflammatory fibrous hyperplasia and irritation fibroma [61]. This was attributed to the presence of elevated levels of estrogen and progesterone in the periodontal tissues. Estrogen leads to increased proliferation of gingival fibroblasts and gingival inflammation [62].

VEGF, commonly upregulated in cancer to promote angiogenesis and sustain tumor growth, is also elevated in benign conditions such as PG, although with clear differences. In PG, VEGF-driven vascular proliferation is localized and benign, lacking the invasive and metastatic potential seen in cancer [63]. The regulatory mechanisms behind VEGF expression in PG are not as well understood as in cancer, where genetic mutations often drive its persistent elevation [63,64]. Treatment responses also diverge; PG usually resolves with local management and VEGF levels normalize, while cancer may require systemic anti-VEGF therapies to control tumor progression [64]. Therefore, while VEGF is a common factor in both PG and cancerous growths, its role and the clinical management of its effects are markedly different, reflecting the benign versus malignant nature of these conditions.

Pyogenic granuloma (PG) is considered a neovascular hyperplastic response in which inducible nitric oxide synthase expression, increased VEGF expression, and low apoptotic rate expression of Bax/Bcl-2 proteins have been reported [39]. In this regard, Blackwell et al. [65] demonstrated the expressions of embryonic stem cell markers, i.e., OCT4, SOX2, pSTAT3 and NANOG, suggesting that the endothelium of PG shows a primitive phenotype. Chen et al. (2008) found overexpression of phosphorylated (p)-activating transcription factor-2 (p-ATF2) and phosphorylated (p)-signal transducer and activator of transcription-3 (p-STAT3) in cutaneous angiosarcoma and PG. Additionally, activation of the MAPK/ERK pathway demonstrated by immunohistochemical evidence for phospho-ERK1/2 positivity, has been identified in oral PG endothelial cells (Figure 3). However, further studies are needed to elucidate the mechanism involved [66].

Investigation of the genetic and genomic alterations associated with PG could provide valuable information on its etiology and pathogenesis. Whole-genome sequencing and comparative genomic analyses could uncover specific genetic mutations or alterations [67].

Recently, *HRAS* mutations were identified in cutaneous PG with lobular capillary arrangements. Subsequently, *KRAS* and *BRAF* mutations were also detected in sporadic cutaneous PG arguing against its classification as a reactive hyperplasia [68,69]. These genes are key components in the mitogen-activated protein kinases (MAPK)/extracellular signal-regulated kinase (ERK) signaling pathway [70].

In a study carried out by Pereira et al., in which they evaluated hotspot mutations in the *HRAS*, *KRAS*, *NRAS*, *BRAF*, *GNA11*, and *GNA14* genes to determine if oral PG shares the same genetic alterations reported in cutaneous PG, representing the same disease in a different topography, they concluded that MAPK/ERK pathway activation in oral PG is not driven by mutations in the *HRAS*, *KRAS*, *NRAS*, *BRAF*, *GNA11* and *GNA14* genes as is PG in skin [66].

## 4. Clinical Features

Pyogenic granuloma (PG) occurs most often in the skin or in the oral cavity, but rarely in the gastrointestinal tract, trachea, urinary bladder, and central nervous system [26,71]. In the oral cavity, the gingiva accounts for 75% of the sites of predilection of this pathology. However, PG may occur in other areas such as the lips, tongue, buccal mucosa, hard plate and peri-implant mucosa, and it affects the maxilla more than the mandible, the anterior region more than the posterior region, with the buccal surfaces more affected than the lingual surfaces [2,3,4]. In the literature, the floor of the mouth is not considered as a site of occurrence of PG. This is perhaps due to the fact that in addition to the absence of sufficient amounts of connective tissue in the mucosa of the floor of the mouth, the tongue protects this region from traumatic injury [72].

Oral PG is a pathology that manifests as a raised, smooth or exophytic growth on a sessile or pedunculated broad base with a smooth and lobulated surface covered with red hemorrhagic and erythematous compressible papules which appear lobulated and warty, complete with ulcerations and covered by a yellow brackish membrane [2,4,10,26]. The surface of the pathology is frequently ulcerated in areas subjected to trauma, and due to its pronounced vascularity, occasional bleeding may occur, especially during mastication. The clinical course of PG is generally slow, asymptomatic and painless [2,26]. The growth of PG is slow, and it takes from weeks to months to reach optimal size [10,26]. As shown in Table 1, the size of PG may vary in diameter from a few millimeters to several centimeters [2,10,26]. The color of this pathology depends on its age: younger PGs tend to be reddish due to the large number of blood vessels, while older ones appear pink in color. The consistency of the oral PG depends on the age of the lesion: as the lesion matures, collagen fibers increase in number, and the lesion becomes firm [15,26].

The signs and/or symptoms referred to by the PG patients in anamnesis are bleeding [73,74,75,76,77,78,79,80,81], difficulty in chewing [2,74,75,76,80,82,83,84], and pain and tenderness [85].

**Table 1 ijms-24-16885-t001:** Reviewed articles.

Year	Author	Country	Age (Years)	Gender	Location	Consistency	Comorbidity	Radiographic	Lesion Size	Diagnosis	Treatment
								Features			
2001	Akyol et al., 2001 [72]	Turkey	4 m	B	Tongue	Soft	No	NA	1 × 0.8 × 0.8 cm	H	Surgical excision
2002	Aguilo L. 2002 [81]	Spain	19 m	B	Gingiva	Soft	No	Fracture of the crown of 61	NA	H	Surgical excision
2006	Parisi E et al., 2006 [6]	USA	33	F	Gingiva	NA	NA	NA	0.5 × 0.5 cm	H	Intralesional corticosteroids
2006	Patil et al., 2006 [86]	India	50	F	Lower Lip	NA	No	NA	1 × 0.5 cm	H	Surgical excision
2006	Shenoy S. 2006 [87]	India	8	G	Gingiva	NA	NA	loss of alveolar crestal bone	2.0 × 1.0 × 1.0 cm	H	Surgical excision
2008	Amirchaghmaghi et al., 2008 [88]	Iran	16	M	Hard palate	Firm	No	NA	0.7 cm	H	Surgical excision
2009	Goncalves 2009 [79]	Brazil	12	F	Upper lip	NA	NA	NA	1.0 cm	H	Surgical excision
2009	Olmedo D et al., 2009 [89]	Argentina	75	F	Peri-implant mucosa	NA	No	No abnormalities	1.0 × 1.0 × 0.6 cm	H	Surgical excision
		Argentina	64	F	Peri-implant mucosa	Firm	No	Bone loss	0.6 × 0.5 × 0.4 cm	H	Surgical excision
2010	Gondivkar et al., 2010 [90]	India	25	F	Gingiva	Soft	Pregnancy	Alveolar bone loss	3 × 7 cm	H	Surgical excision
2010	Rizwanulla et al., 2010 [91]	Nepal	13	F	Gingiva	Firm	No	NA	2.0 × 1.0 × 1.0 cm	H	Surgical excision
2011	Behl et al., 2011 [85]	India	60	F	Gingiva	Firm	No	Vertical bone loss	3.2 × 3.4 cm	H	Surgical excision
2011	Mubben et al., 2011 [76]	India	63	F	Gingiva	Soft	No	NA	5.0 × 3.5 cm	H	Surgical excision
2011	Penseriya et al. 2011 [82]	India	30	M	Gingiva	NA	No	Interdental bone loss	2.0 × 1.5 cm	H	Surgical excision
2011	Shivaswamy et al., 2011 [92]	India	19	M	Gingiva and palate	Soft	NA	Horizontal bone loss	4.0 × 5.0 mm	H	Surgical excision
2012	Chandrashekar 2012 [93]	India	28	F	Gingiva	Soft	No	NA	NA	H	Surgical excision
2012	Panjwani et al., 2012 [75]	India	69	M	Tongue	Firm	No	NA	2.0 × 3.0 cm	H	Surgical excision
2012	Ravi et al., 2012 [94]	India	33	M	Lower lip	Firm	NA	NA	3.0 × 2.0 cm	H	Surgical excision
2012	Piscoya et al., 2012 [95]	Brazil	44	M	Lower lip	NA	NA	NA	NA	H	Surgical excision
2012	Verma et al., 2012 [96]	India	30	F	Gingiva	NA	No	Alveolar bone loss	1.5 × 1.0 cm	H	Surgical excision
2013	Adusumilli et al., 2013 [97]	India	24	F	Gingiva	Firm	NA	NA	2 × 3.5 cm	H	Surgical excision
			27	F	Gingiva	Firm	NA	NA	2 × 3.5 cm		Surgical excision
			27	F	Gingiva	Firm	NA	NA	2 × 3.5 cm	H	Surgical excision
			32	F	Gingiva	Firm	NA	NA	1.5 × 2.5 cm	H	Surgical excision
			23	F	Gingiva	Soft	NA	NA	1.5 × 1.75 cm	H	Surgical excision
			26	F	Gingiva	Soft	NA	NA	2.0 × 2.5 cm	H	Surgical excision
			28	F	Gingiva	Soft	NA	NA	0.5 × 0.8 cm	H	Surgical excision
2013	Deshmukh et al., 2013 [83]	India	9	M	Gingiva and mucogingival junction	Soft	No	Horizontal bone loss	NA	H	Surgical excision
2013	Gomes et al., 2013 [2]	India	22	M	Gingiva	NA	NA	No bone loss	2.1 × 4.4 cm	H	Surgical excision
2013	Kamala 2013 [3]	India	30	F	Upper lip	Firm	No	NA	0.8 cm	H	Surgical excision
2013	Mahabob et al., 2013 [98]	India	22	F	Palate	Firm	Pregnancy	NA	2 × 2 cm	H	Surgical excision
2013	Moraes, et al., 2013 [99]	Brazil	65	M	Gingiva	NA	Diabetes and high blood pressure	No abnormalities	NA	H	Surgical excision
2013	Sangamesh et al., 2013 [100]	India	40	F	Buccal mucosa	Firm	NA	NA	1.5 × 1.5 cm	H	Surgical excision
2014	Asha et al., 2014 [84]	India	54	M	Lower lip	Firm	No	NA	3.0 × 3.0 cm	H	Surgical excision
2014	Fekrazad et al., 2014 [96]	Iran	24	F	Gingiva	Soft	NA	No abnormalities	1.4 × 0.8 mm	H	Laser excision
2014	Ghalayani et al., 2014 [5]	Iran	45	M	Tongue	NA	Epilepsy beginning	NA	4.0 × 3.0 × 1.0 cm	H	Surgical excision
2014	Kejriwal et al., 2014 [101]	India	59	M	Gingiva	Firm	No	No abnormalities	1.5 × 2.0 × 1.0 cm	H	Surgical excision
2014	Mastammanavar et al., 2014 [77]	India	44	F	Gingiva	Soft	NA	NA	1.5 × 3.0 cm	H	Surgical excision
2014	Sun et al., 2014 [102]	China	22	F	Gingiva	NA	Pregnancy	NA	3 cm	C	No
2015	Asnaashari et al., 2015 [103]	Iran	6	M	Gingiva	Soft	No	No abnormalities	1.1 × 1.3 cm	H	Laser excision
2015	Bugshan et al., 2015 [104]	USA	51	F	Gingiva and palatal	NA	NA	NA	0.9 × 0.6 cm buccal and 0.8 × 0.7 cm palatal.	H	Intralesional injections
2015	De Carvalho et al., 2015 [105]	Brazil	11	B	Upper Lip	NA	No	NA	4.5 cm	H	Surgical
2015	Ganesan A. 2015 [80]	India	49	F	Gingiva	Firm	No	No bone loss	4.0 × 3.0 cm	H	Surgical excision
2015	Sachdeva 2015 [106]	India	45	F	Buccal mucosa	Firm	No	NA	2 × 1 cm	H	Surgical excision
2015	Tripathi et al., 2015 [107]	India	55	M	Gingiva	NA	No	Alveolar bone loss	3 × 3 cm	H	Surgical excision
2016	Agarwal N et al., 2016 [108]	India	8 d	B	Gingiva	Soft	No	NA	0.5 × 0.8 cm	H	Surgical excision
2016	Al-Mohaya et al., 2016 [22]	Saudí arabia	51	F	Gingiva	Firm	Uncontrolled type II diabetes mellitus	NA	2.0 × 1.5 cm	H	Laser excision
2016	Marla et al., 2016 [109]	Nepal	40	F	Gingiva	NA	NA	NA	1.0–2.0 cm	H	Surgical excision
		Nepal	40	F	Buccal mucosa	NA	NA	NA	1.0–2.0 cm	H	Surgical excision
		Nepal	9	M	Buccal mucosa	NA	NA	NA	<1.0 cm	H	Surgical excision
		Nepal	23	F	Gingiva	NA	NA	NA	1.0–2.0 cm	H	Surgical excision
2016	Cheney et al., 2016 [21]	USA	16	M	Tongue, Buccal mucosa,	Soft	Acute lymphoblastic leukemia	N/A	N/A		Surgical excision
		USA	14	G	Tongue	N/A	Fanconi anemia	No	0.5 × 0.2cm 1.0 × 0.3 cm 0.3 × 0.2	H	Surgical excision
		USA	11	G	Tongue	N/A	Fanconi anemia	No	2.0 × 2.5 cm	H	Surgical excision
		USA	15		Buccal mucosa		Stage IIIB nodular sclerosing Hodgkin lymphoma.	No	1 cm 2.0 × 4.0 cm	H	CO2 laser
		USA	6	B	Tongue	N/A	Junctional epidermolysis bullosa	No	1.0 × 0.5 cm	H	Surgical excision
2017	Rosa et al., 2017 [73]	México	34	F	Gingiva	Firm	No	Absence of interproximal contact	1.5 × 0.9 cm	H	Surgical excision
2018	Tenore et al., 2018 [110]	Italy	28	F	Retromolar region	Soft	No	No abnormalities	NA	H	CO_2_ laser
			58	M	Gingiva	Soft	No	No abnormalities		H	CO_2_ laser
			19	M	Retromolar region	Soft	No	No abnormalities	NA	H	CO_2_ laser
			21	M	Retromolar region	Soft	No	No abnormalities	NA	H	CO_2_ laser
2018	Canivell et al., 2018 [111]	Spain	32	F	Lower lip	NA	Pregnancy	NA	0.5 × 1.0 cm	H	Surgical excision
2018	Parajuli et al., 2018 [74]	Nepal	26	F	Tongue	Soft	No	NA	2.5 × 2.0 cm	H	Surgical excision
		Nepal	15	G	Upper lip	NA	NA	NA	0.5 × 0.5 cm	C	Surgical excision
2019	Poudel et al., 2019 [112]	Nepal	49	M	Upper lip	Firm	NA	NA	0.6 × 0.8 cm	H	Surgical excision
2020	Gutierrez 2020 [24]	Perú	51	F	Alveolar ridge	NA	Hypothyroidism erythrodermic psoriasis	Alveolar bone loss	2.5 × 2.5 cm	H	Surgical excision
2020	Banjar et al., 2020 [113]	Saudí arabia	15	B	Lower lip	Soft	No	NA	1.2 × 0.8 × 0.6 cm	H	Surgical excision
2021	Aragaki et al., 2021 [27]	Japan	55	M	Tongue	Soft	Gastric Cancer	No abnormalities	0.6cm	H	Surgical excision
			67	M	Upper lip	Soft	Gastric Cancer	No abnormalities	0.5 mm	H	Surgical excision
2021	Pisano et al., 2021 [114]	Italy	11	G	Lower lip	Soft	No	N/A	1.5 cm	H	Diode Laser
2023	Lomelí et al., 2023 [115]	México	32	F	Hard palate	Soft	No	Alveolar bone loss	25 × 12 mm	H	Surgical excision
		México	42	F	Gingiva	Soft	No	Alveolar bone loss	16 × 10 mm	H	Surgical excision
			38	F	Gingiva				20 × 15 mm		

Note: (m): months. (d): Day. (F): Female. (M): Male. (B): Boy. (G): Girl. (NA): data not available. (H): Histopathological. (C): Clinical.

## 5. Radiographic Features

Oral PG generally does not present radiographic findings: some authors make a presumptive diagnosis with the clinical features. Thus, most of the clinical cases described in the literature do not present radiographic analysis and/or description. However, some authors have reported bone loss or erosion of the alveolar ridge associated with the area where the PG is located [73,80,83,85,87,90,92,96,107,115] (Table 1).

The molecular mechanisms leading to bone loss or erosion associated with oral PG are unclear. However, PG is a benign inflammatory lesion that expresses significantly more VEGFs and basic fibroblast growth factors than healthy gingiva and periodontitis [37]. Growth factors such as fibroblast growth factor-2, growth arrest-specific gene 6, and TNF-α, among other molecules, stimulate mature osteoclast function and survival through activation of extracellular signal-regulated kinase (ERK), resulting in degradation or resorption of organic and inorganic bone components [17]. The ERK signaling pathway has been associated with the regulation of osteoclasts with respect to survival, proliferation, apoptosis, formation, polarity, podosome disassembly, and differentiation [17].

This signaling cascade constitutes the core of three serially phosphorylated protein kinases. Activation of Raf isoforms through the Ras–Raf interaction stimulates MAPKK, MEK1 and MEK2, and then activates ERK1 and ERK2 via dual phosphorylation at the conserved Thr–Glu–Tyr (TEY) motif, leading to phosphorylation of various downstream substrates [17,116,117]. Among the latter are c-Fos, NFATc1, MITF, TFE3, Hedgehog-Gli, Egr2, RSK2 and MMp-9, which ultimately activate the ERK signaling pathway [17]. Pereira et al. [66] reported MAPK/ERK pathway activation, as demonstrated by the immunohistochemical positivity of phospho-ERK1/2 in oral PG endothelial cells. Future research should focus on elucidating the molecular mechanisms that trigger PG-associated bone loss or erosion, particularly the role of the MAPK/ERK pathway in this process (Figure 3).

## 6. Microscopic Features

Most PGs are constituted by a lobular mass of hyperplastic granulation tissue representing a non-neoplastic proliferation formed by abundant vascular spaces, young fibroblasts, and infiltration of acute and/or chronic inflammatory cells with lymphocytes, plasma cells, histiocytes, and polymorphonuclear cells in a collagen matrix. This entity is partially or totally covered by parakeratotic or non-keratinized stratified squamous epithelium. Ulcerated lesions exhibit an extensive fibrin layer in the epithelium [2,26].

## 7. Differential Diagnosis of Oral PG

The features used for differential diagnosis of oral PG include red or reddish-blue growths such as gingival hyperplasia, peripheral giant cell granuloma, hemangiomas, conventional granulation tissue, peripheral ossifying fibroma, peripheral odontogenic fibroma, metastatic cancer, Kaposi’s sarcoma, bacillary angiomatosis, angiosarcoma, and non-Hodgkin’s lymphoma [2,3,4,10,26].

## 8. Treatment of Oral PG

The treatment or the management of oral PG depends on the particular characteristics presented by each patient. However, the treatment of choice is conventional surgical excision. Other minimally invasive treatment modalities have been suggested, including laser, corticosteroid injections, cryosurgery and sclerotherapy [2,4,26] (Table 1).

Surgical excision consists of the complete removal of the lesion and the extension of the cut to the periosteum, including a 2 mm margin to the adjacent soft tissues. If the PG is located near adjacent teeth, it is important that after removal of the lesion, debridement is performed both supra- and sub-gingivally to the biofilm and/or dental calculus. Additionally, it is important to remove all irritating agents (foreign materials, sources of trauma, overhang crowns, etc.) that are present in the area of the lesion. These suggested measures, both in the surgical technique and in the removal of irritants, are aimed at avoiding recurrence of PG [2,7,73,115].

Chandrashekar implemented a minimally invasive approach as a treatment strategy for oral PG. This protocol consists of performing scaling and root planning in the area where the lesion is located. In addition, it is crucial to maintain complete oral hygiene by brushing twice a day and using a 0.12% chlorhexidine rinse twice a day. It is necessary to monitor the evolution of the lesion every week. If the lesion persists, scaling and root planning should be implemented every week for four consecutive weeks in order to continue with the non-invasive approach. At the same time, it is recommended that patients should maintain adequate brushing and flossing twice a day. This minimally invasive treatment may be considered when the PG is small in size, painless, and without bleeding [93].

Recently, a laser-assisted removal treatment was used for oral PG [79,114,118]. Asnaashari et al. [103] implemented the Er:YAG Laser therapy using the Diode Laser Gallium–Aluminum–Arsenide (GA–LA–AS). Moreover, Al-Mohaya et al. [118] used the 940 nm diode laser in an uncontrolled diabetic patient without observing recurrences and/or postoperative complications. Additionally, Tenore et al. [110] presented four cases of PG in patients with a history of excision of oral neoplastic lesions and rehabilitation by a free revascularized flap of the iliac crest; excisional biopsies were performed by carbon dioxide (CO_2_) laser. These cases presented, suggest that limitations in oral functions and maintenance of oral hygiene measures subsequent to reconstruction surgery with revascularized free flap probably played a role in the development of gingival reactive hyperplastic lesions with the presence of triggering factors such as local trauma, chronic infection or inadequate prosthesis. The advantages of applying the laser are that by sealing the blood vessels and nerve bundles, there is no bleeding during surgery, thereby ensuring better visualization of the surgery site, sterile conditions, cutting precision, reduced number of instruments used, and a suture-less procedure with minimal postoperative pain [114,118]. In addition, the laser instantly disinfects the surgical wound, resulting in lower possibility of postoperative infection, minimal inflammation, and better healing of the surgical wound. In oral cavity lesions, the use of a laser reduces intraoperative and postoperative complications, when compared to surgical excision [118].

A proposed therapeutic alternative is corticosteroid injection into the oral PG, an option which was identified in two studies in this review (Table 1). Parisi et al. [6] were pioneers in implementing, for the first time, the use of a series of corticosteroid injections into lesions for the management of multiple intraoral PG nodules as a new conservative treatment option for this pathology. However, Bugshan et al. [104] suggested a series of injections in five different sites of the lesion with 0.1 mL of 10 mg/mL triamcinolone acetonide, without exceeding a total of 0.5 mL, in addition to local application of 0.05% clobetasol propionate for 2 weeks. The latter protocol is appropriate and effective, particularly in patients with high PG recurrences due to poor surgical excisions. The exact mechanism of action of corticosteroid therapy is still unknown. However, these drugs may improve the response of the lesion in the vascular bed to vasoconstrictor agents. A corticosteroid such as dexamethasone, stops vascular proliferation by downregulating proangiogenic factors such as VEGF-A, MMP1, and IL-6 [9].

Cryotherapy is a simple, easy-to-execute, inexpensive, and safe treatment that has become one of the therapeutic options used for PG patients. Its implementation is simpler than that of surgical excision, and it is cheaper than laser. It allows resolution of the pathology without leaving significant scarring. Endothelial cells may be more vulnerable to cryotherapy than collagen fibers [119].

Sodium tetradecyl sulfate sclerotherapy is among the alternative therapies that have been implemented. This technique offers a better alternative to excision due to its simplicity and the absence of scarring, although multiple treatment sessions are required. The therapeutic effects of this treatment may be mediated through a mechanism involving the specific and non-specific actions of sodium tetradecyl sulfate, which specifically causes damage to endothelial cells and obliterates the lumen of vessels. Additionally, in stromal tissues, it may cause non-specific necrotic changes. The adverse effects of treatment with sodium tetradecyl sulfate include allergic reactions, skin necrosis, and hyperpigmentation. Sodium tetradecyl sulfate injection is usually painless. Therefore, extravasation may develop asymptomatically. To avoid skin necrosis, slow and careful injection with some pressure is necessary [120].

During pregnancy, choosing the treatment modality for oral PG is challenging, and it depends on the severity of the symptoms. Gondivkar et al. [90] and Canivell et al. [111] have proposed surgical excision of the granuloma during pregnancy. On the other hand, Su et al. [102] suggest that for patients without hemorrhagic or painless lesions, oral hygiene instructions, clinical monitoring of development of PG, follow-up, and oral self-care at home should be implemented. Regardless of the treatment implemented for PG in pregnant women, we suggest, particularly in this group of patients, a strict control of oral hygiene, the continuous removal of biofilm and/or dental calculus, the use of soft toothbrushes, as well as flossing in order to avoid recurrence.

The formation of a reactive lesion (for example, PG) around dental implants is a complication that has received attention. These lesions may lead to marginal bone loss and, consequently, implant failure. Therefore, several treatments for implant-associated PG have been considered, with the treatment of choice being conservative surgical excision with total removal of the base of the lesion and bone curettage [121]. Additionally, it is recommended that the implant surface should be polished so as to eliminate any irritating factor that may be causing or favoring the appearance of PG [121]. Photodynamic therapy is a simple, non-invasive adjunctive treatment for peri-implant diseases. It allows for control of disease progression through de-contamination of infected surfaces. However, further studies are needed to establish its efficacy in cases of reactive lesions around dental implants [121,122].

Regarding postoperative management for arresting recurrence of PG after treatment, some authors have suggested implementation of oral hygiene measures, among which are the use of a soft toothbrush, keeping the lesion area clean, use of mouth rinses, and antibiotic therapy [73,85,92,103]. The suggested rinses entail the use of 0.2% chlorhexidine [79,92,101] and 0.12% chlorhexidine [73]; use of gluconate twice daily, and a saline rinse [115]. The antibiotic of choice frequently used for adequate intervention is amoxicillin at a dose of 500 mg every 8 h for 5 days [75,76,77]. In pediatric patients, good oral hygiene is recommended for keeping the lesion area clean, and neither analgesics nor antibiotics are prescribed [103,114].

The recurrence rate of PG is 15%, and this is associated with incomplete lesion removal and failure to eliminate etiological factors [13,101]. Frumkin et al. [123] proposed a conservative protocol for preventing recurrence. This entails removal of irritants with debridement under local anesthesia, along with ancillary measures such as chlorhexidine rinse and oral hygiene instructions. In addition, the protocol suggests a follow-up schedule that involves visits every 2 weeks during the first 2 months, followed by maintenance visits once every 2 months. However, recurrence is uncommon in extra-gingival locations after surgical excision [123].

## 9. Future Perspective

Future research on PG is aimed at making significant advances by delving into the molecular genesis of the lesion, seeking novel and minimally invasive therapeutic approaches, and understanding the complexities behind its tendency to recur. A concerted effort to analyze the hormonal interaction at work, together with the search for reliable biomarkers for diagnosis and prognosis, will shed new light on the clinical trajectory of PG. Adopting cutting-edge diagnostic technologies, gathering extensive epidemiological knowledge, and conducting rigorous clinical trials will be critical to refining treatment regimens. Additionally, the focus on refining patient management strategies and developing comprehensive, standardized treatment protocols is expected to improve. Notably for the quality of life and clinical outcomes of people fighting this enigmatic injury.

## 10. Conclusions

Pyogenic granuloma (PG) manifests as a benign proliferation of connective tissue, primarily characterized by granulation tissue hyperplasia. This phenomenon often occurs on the skin or mucous membranes and is considered a reactive process triggered by various factors such as trauma, dental calculi, chronic irritation, the use of certain drugs, and is linked to elevated levels of estrogen and progesterone. In its pathogenesis, intense inflammatory activity stands out, facilitated by the imbalance between enhancers and inhibitors of angiogenesis, as well as interactions between mast cells and fibroblasts. Exploring the entire genome and comparative genomic analyses could reveal specific mutations or genetic alterations associated with the underlying molecular mechanisms of PG. This approach could uncover potential innovative and less invasive therapeutic pathways.

## Figures and Tables

**Figure 1 ijms-24-16885-f001:**
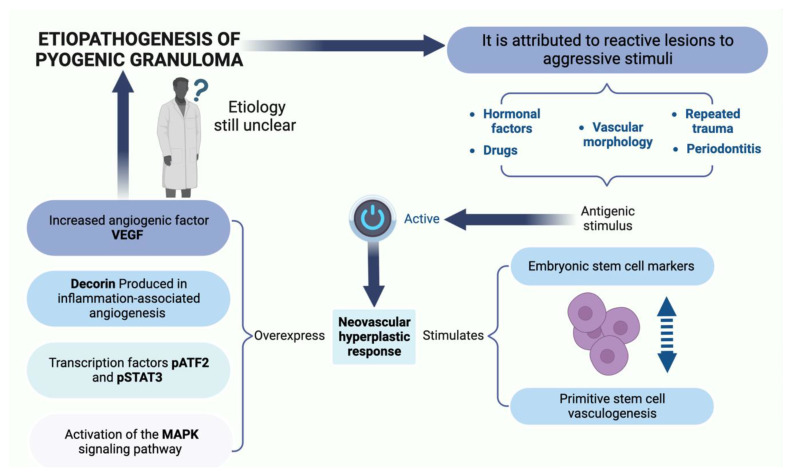
The etiopathogenesis and risk factors of PG observed. Figure created with BioRender, © biorender.com.

**Figure 2 ijms-24-16885-f002:**
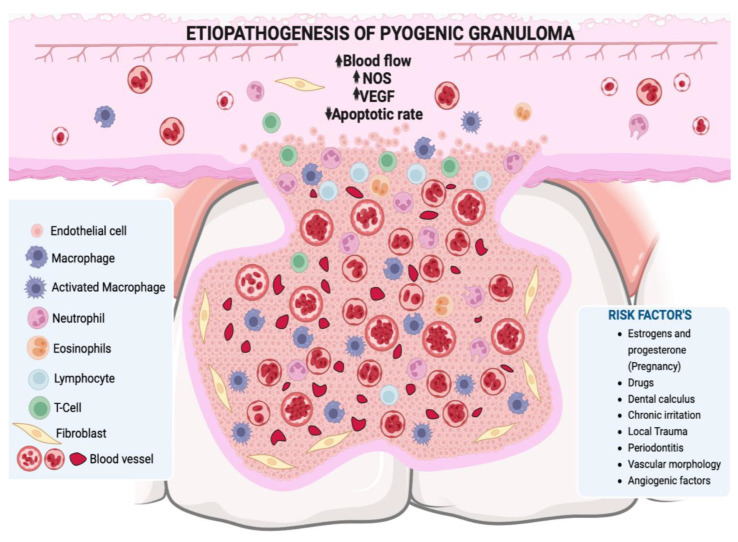
A lobular mass of hyperplastic granulation tissue is observed, formed by abundant vascular spaces, young fibroblasts, infiltration of acute and/or chronic inflammatory cells with macrophages, neutrophils, lymphocytes, and polymorphonuclear cells. NOS: nitric oxide synthase, VEGF: vascular endothelium-derived grow. Figure created with BioRender, © biorender.com.

**Figure 3 ijms-24-16885-f003:**
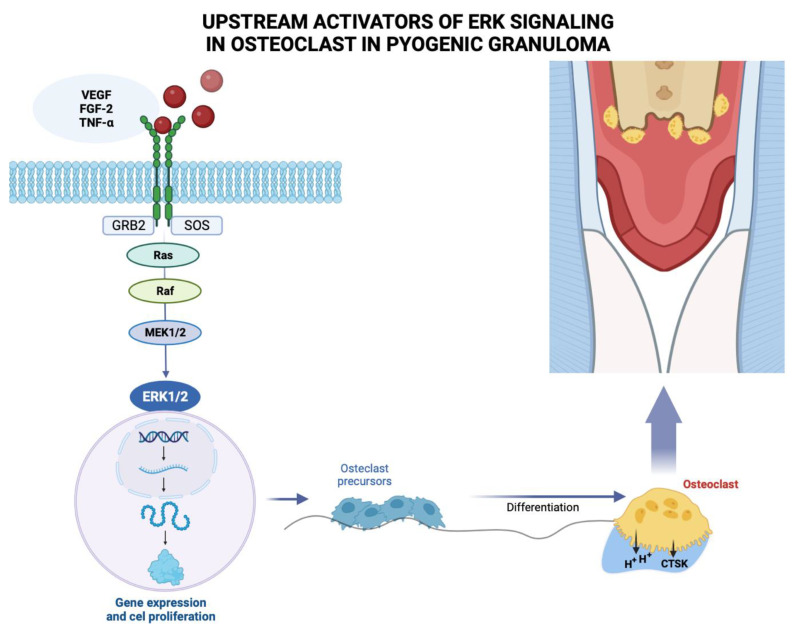
Growth factors such as vascular endothelium-derived growth factor (VEGF), fibroblast growth factor-2 (FGF-2), tumor necrosis factor α (TNF-α) among other molecules, bind to the extracellular domain of a receptor tyrosine kinase that acts with docking sites for GRB2 adapter proteins that also binds to a guanine nucleotide exchange factor SOS, for a protein called Ras, which activates another signaling protein called Raf, acting on the MEK substrate 1/2, which phosphorylates and activates ERK 1/2, which enters the nucleus where it activates specific transcription factors and genes involved in the proliferation of osteoclast precursors that differentiate into mature osteoclasts that will resorb the bone crest in the pyogenic granuloma area. Figure created with BioRender, © biorender.com.

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
