# Peer review of "Oral Pyogenic Granuloma: A Narrative Review"

_ijms, 2023, doi:10.3390/ijms242316885_

Round 1
Reviewer 1 Report
Comments and Suggestions for Authors
For a submission targeting IJMS, ¾ of the text concerns a description of this pathology such as we give to medical or dental students. This text would probably be better suited to a Journal of Clinical Medicine-style journal.
If the authors maintain their wish to submit to IJMS, my proposal would be to focus on the "3. Etiopathogenesis" section and develop it further.
Author Response
Thank you very much for taking the time to review this manuscript. Please find the detailed responses below and the corresponding revisions/corrections highlighted/in track changes in the re-submitted files.

Reviewer 2 Report
Comments and Suggestions for Authors
This is a well-written review of oral pyrogenic granuloma. The author addressed it well but some of my concerns are,
1. The abstract needs to be clearer and more oriented.
2. Epidemiology needs to be clear (lines 64-68)
3. I suggest a more critical review exploring the pathology with oral inflammatory pathways. Please give more attention to the basic science papers that demonstrate potential links.
4. Vegf will upregulate in most cancerous growth how it can differ in PG
5. The conclusion should be elaborated with the latest references.
6. Future directions are also needed in a separate topic.
7. Reference sections should be updated with more latest references
Comments on the Quality of English LanguageModerate editing of English language is required
Author Response

(The authors gave the same response as above.)

Reviewer 3 Report
Comments and Suggestions for Authors
The manuscript is well presented. Some issues have been observed;
- The study should be described in the title as "a narrative review" rather than "a literature review" to avoid confusion for the reader. In particular, the authors did not give any data about the approach to the literature databases.
- In the figures, there are these words "created in biorender.com". The authors should confirm the originality of the figures and any subsequent licenses for publishing these figures.
- In the literature, there is a reported experience of oral pyogenic granuloma in orally Rehabilitated Patients by Free Revascularized Flap, it is preferable to discuss this issue to make the study more comprehensive. You can see this paper “Tenore et al. Gingival Reactive Lesions in Orally Rehabilitated Patients by Free Revascularized Flap. Case Rep Dent. 2018”.
Author Response

(The authors gave the same response as above.)

Round 2
Reviewer 3 Report
Comments and Suggestions for Authors
The paper has been improved.